# Normalized Conditional Mutual Information Surrogate Loss for Deep Learning Classifiers

**Linfeng Ye[§‡], Zhixiang Chi[‡]**
[§]University of Waterloo, Waterloo, Canada    [‡]University of Toronto, Toronto, Canada

## Abstract

In this paper, we propose a novel information-theoretic surrogate loss, dubbed normalized conditional mutual information (NCMI), as a drop-in alternative to the de facto cross-entropy (CE) for training deep neural networks (DNNs)-based classifiers. We first observe that the DNN's NCMI is inversely proportional to its accuracy. Building on this insight, we introduce an alternating algorithm to efficiently minimize the NCMI to train the DNN-based classifiers. Geometrically, NCMI train the DNN by encouraging the intra-class concentration and inter-class separation. Across image recognition benchmarks, NCMI-trained models surpass state-of-the-art losses by substantial margins at a computational cost comparable to that of CE. Notably, on ImageNet, NCMI yields a 2.77% top-1 accuracy improvement with ResNet-50 compared to the CE. Gains are consistent across various architectures and batch sizes, suggesting that NCMI is a practical and competitive alternative to CE. All code and data are publicly available at `code`.

## 1 Introduction

Cross-entropy (CE), first introduced by Cox (1958) for binary classification in the 1950s as the objective function for analyzing binary sequences, remains the dominant surrogate loss for training deep neural classifiers. Despite its empirical success, its capability for multi-class classification has not been justified until very recently (Yang et al., 2025). CE primarily enforces agreement with one-hot targets by minimizing the negative log-likelihood. A broad literature further proposes CE-alternatives to improve the model's classification accuracy; we review them in Section 2.

From the information geometry perspective, the learning process of most previous approaches can be regarded as optimizing the model so that the output clusters are pulled toward predefined distributions on the probability simplex. In this work, we ask a different question:

*Can we train classifiers by shaping the output distribution to be (i) concentrated within each class and (ii) well separated across classes, using an information-theoretic principle?*

Following Yang et al. (2024), we model the DNN-based classification task as a three-state Markov chain Cover (1999), as illustrated in Figure 1. We quantify the concentration and separation of the output distribution using the conditional mutual information (CMI) and $\Gamma$, respectively (see Section 4). We then define the normalized conditional mutual information (NCMI) as the ratio between the CMI and $\Gamma$. We observe that model's classification accuracy is inverse proportional to its NCMI value. Motivate by this observation, instead of maximizing the log-likelihood (i.e., minimizing CE), we train the model to minimize its NCMI. After training, we evaluate the model using centroid-based decisions by comparing its outputs to class centroids. Extensive experiments on CIFAR-100 and ImageNet show that NCMI-trained models consistently outperform strong CE-based baselines at a computational cost comparable to CE. The key contributions are as follows:

• We propose a new CE alternative named NCMI for training DNN-based classifiers.

• To minimize the NCMI, we introduce a novel alternating optimization algorithm that minimizes the NCMI loss.

• To evaluate the NCMI's effectiveness, we conduct comprehensive experiments on two natural-image datasets, namely, CIFAR-100 (Krizhevsky et al., 2009) ImageNet (Deng et al., 2009). Although modern DNNs and optimization tricks are tailored to CE–based surrogates, NCMI achieves state-of-the-art classification performance across all benchmarks.

## 2 RELATED WORK

Within the existing literature, CE and its variants are the de facto objectives for DNN-based classifier training. Several works have attempted to augment CE with various regularizers. Hui et al. (2023) add an $\ell_2$ penalty to the non-ground-truth entries of the predicted probability distribution, and OPL (Ranasinghe et al., 2021) explicitly clusters same-class features while enforcing orthogonality between different classes in the penultimate layer.

Another line of work improves classification accuracy by modifying CE. Focal Loss (Lin et al., 2017) down-weights well-classified examples via a power transformation so training emphasizes hard instances. PolyLoss (Leng et al., 2022) reframes standard classification losses as polynomial expansions. Hui & Belkin (2021) propose SquareLoss and empirically find that mean square loss performs on par with or even outperforms CE on modern DNNs. Su-

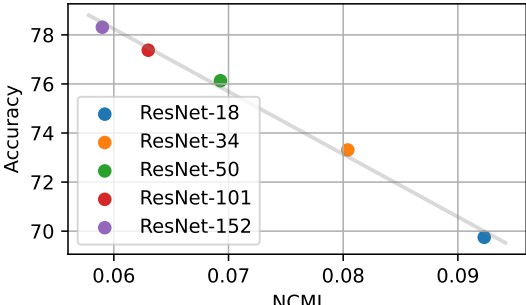

Figure 1: Mappings from the label space $Y$ to the input space $X$, and from the input space to an output space $\hat{Y}$. Input $\boldsymbol{x}$ is sampled from the class $Y = y$ according to the $P_{X|Y}(\cdot|y)$. This is further mapped by a DNN and a simplex-valued function to an output probability distribution $\boldsymbol{p} \in \mathcal{P}$.

Figure 2: The accuracy vs NCMI value over the validation set of pretrained ResNet models on the ImageNet dataset.

pervised contrastive learning (SupCon) (Khosla et al., 2020) pulls together same-class embeddings and pushes apart different-class embeddings, followed by a linear classifier trained on the frozen features. AntiClass (Katsikas et al., 2025) replaces the one-hot target with a one-cold target to mitigate neural collapse.

In contrast, we study the surrogate loss for classification through the lens of information geometry. NCMI trains the DNN by encouraging the intra-class concentration and inter-class separation of the output distribution cluster.

## 3 NOTATION

Scalars are denoted by non-bold letters (*e.g.* $\beta$), For a positive integer $C$, let $[C] \triangleq \{1, \ldots, C\}$, vectors by bold lowercase letters (*e.g.* $\boldsymbol{a}$), the $i$-th entry of a vector $\boldsymbol{a}$ is written as $\boldsymbol{a}[i]$. We denote $\mathcal{P}^k$ as the set of all $k$-dimensional probability distributions. For any two probability distributions $\boldsymbol{p}, \boldsymbol{q} \in \mathcal{P}^k$, the Kullback–Leibler (KL) divergence is defined as $D(\boldsymbol{p}\|\boldsymbol{q}) = \sum_{i=1}^{k} \boldsymbol{p}[i] \ln \frac{\boldsymbol{p}[i]}{\boldsymbol{q}[i]}$, where $\ln$ denotes the logarithm with base $e$, and we write the CE of the one-hot probability distribution corresponding to $y$ and $\boldsymbol{q}$ as $H(y, \boldsymbol{q}) = -\ln \boldsymbol{q}_y$. We denote $\mathbb{1}$ as the indicator function. We define the normalized sigmoid function (NSF) $\sigma^{NSF}$ and softmax function $\sigma^{SM}$ as

$$\sigma^{NSF}(\boldsymbol{z})[j] = \frac{\phi(\boldsymbol{z}[j])}{\sum_{i=1}^{n} \phi(\boldsymbol{z}[i])}, \quad \phi(\boldsymbol{z}[i]) = \frac{1}{1 + e^{-\boldsymbol{z}[i]}}; \quad \sigma^{SM}(\boldsymbol{z})[i] = \frac{e^{\boldsymbol{z}[j]}}{\sum_{i=1}^{n} e^{\boldsymbol{z}[i]}}, \text{ where } \boldsymbol{z} \in \mathbb{R}^k.$$
$$(1)$$

Given a multi-class classification dataset $\mathcal{D}$ with $C$ classes, let $\mathcal{D}^y \subseteq \mathcal{D}$ denote the subset of samples labeled $y$, and $\boldsymbol{c}_{\boldsymbol{x}}$ denote the label of sample $\boldsymbol{x}$.

## 4 MODELING CLASSIFICATION AS A MARKOV CHAIN

In a classification task with $C$ classes, a DNN $f$, a linear classifier $h$ and a simplex-valued function $\sigma$ can be regarded as a mapping $(\sigma \circ h \circ f) : \boldsymbol{x} \to \boldsymbol{p}_{\boldsymbol{x}}$, where $\boldsymbol{x}$ is an input, and $\boldsymbol{p}_{\boldsymbol{x}} \in \mathcal{P}^n$ is the output probability distribution. Usually, $n = C$ when we use CE as the surrogate loss. Following Yang et al. (2024; 2025), we model the classification task as a three-state Markov chain, as depicted in Figure 1. We can empirically quantify the concentration of DNN's output by conditional mutual information (CMI) between the model's input and output, given input's ground truth as

$$I(X; \mathcal{P}|Y) = \frac{1}{|\mathcal{D}|} \sum_{y} \sum_{\boldsymbol{x} \in \mathcal{D}^y} D(\boldsymbol{p}_{\boldsymbol{x}}\|\boldsymbol{s}^y), \text{ where } \boldsymbol{s}^y \triangleq \frac{1}{|\mathcal{D}^y|} \sum_{\boldsymbol{x} \in \mathcal{D}^y} \boldsymbol{p}_{\boldsymbol{x}}, \text{ for } y \in Y. \quad (2)$$

Further, the separation of DNN's output can be empirically quantified as

$$\Gamma = \frac{1}{|\mathcal{D}|^2} \sum_{\boldsymbol{z} \in \mathcal{D}} \sum_{\boldsymbol{x} \in \mathcal{D}} \mathbb{1}_{\{c_{\boldsymbol{x}} \neq c_{\boldsymbol{z}}\}} D(\boldsymbol{s}^{c_{\boldsymbol{x}}} \| \boldsymbol{p}_{\boldsymbol{z}}) \tag{3}$$

Ideally, we want $I(X; \mathcal{P}|Y = y)$ to be small while keeping $\Gamma$ large. This leads us to consider the ratio between $I(X; \mathcal{P}|Y = y)$ and $\Gamma$.

$$\hat{I}(X; \mathcal{P}|Y) \triangleq \frac{I(X; \mathcal{P}|Y)}{\Gamma}. \tag{4}$$

We refer to $\hat{I}(X; \mathcal{P}|Y)$ as the normalized conditional mutual information (NCMI). To study its relationship with classification performance, we discard the fully connected classifier of each pretrained ResNet and apply softmax to the penultimate feature vector to obtain a probability distribution, from which we compute NCMI according to Equation (4). For top-1 evaluation, we form class centroids $s^y$ as in Equation (2) and classify each sample by minimum KL-divergence assignment in the induced probability space. Results on the ImageNet validation set are reported in Figure 2. We observe a clear inverse linear relationship: models with lower NCMI achieve higher accuracy, with a Pearson correlation coefficient exceeding $-0.997$. This suggests that, for a fixed DNN family, improving performance is associated with simultaneously reducing both the error rate and the model's NCMI during training. Motivated by this observation, in the next section, we demonstrate that NCMI per se suffices for training DNN classifiers.

## 5    METHODOLOGY

Previous discussions suggest a new surrogate loss for training DNN-based classifiers. Specifically, in the learning process, instead of minimizing the CE, which pulls the output toward predefined distributions, we aim simultaneously encourages the intra-class concentration and inter-class separation by minimizing $\hat{I}(X; \mathcal{P}|Y)$. The optimization problem can be written as

$$\min_{\boldsymbol{\theta}} \hat{I}(X; \mathcal{P}|Y) = \min_{\boldsymbol{\theta}} \frac{\frac{1}{|\mathcal{D}|} \sum_y \sum_{\boldsymbol{x} \in \mathcal{D}^y} D(\boldsymbol{p}_{\boldsymbol{x}} \| \boldsymbol{s}^y)}{\frac{1}{|\mathcal{D}|^2} \sum_{\boldsymbol{z} \in \mathcal{D}} \sum_{\boldsymbol{x} \in \mathcal{D}} \mathbb{1}_{\{c_{\boldsymbol{x}} \neq c_{\boldsymbol{z}}\}} D(\boldsymbol{s}^{c_{\boldsymbol{x}}} \| \boldsymbol{p}_{\boldsymbol{z}})} \tag{5}$$

We note that the objective in Equation (5) is not amenable to parallel computation on GPUs due to the dependency of $\hat{I}(X; \mathcal{P}|Y)$ on the centroid $\boldsymbol{s}^y$ of each cluster corresponding to $Y = y$. To overcome this, we introduce a dummy distribution $\boldsymbol{q}^y \in \mathcal{P}^n$ for each $y \in [C]$ and convert it into a double minimization problem, as shown in the following theorem, which will be proved in the Appendix.

**Theorem 1** *For any DNN:* $\boldsymbol{x} \rightarrow \boldsymbol{p}$,

$$\min_{\boldsymbol{\theta}} \frac{\frac{1}{|\mathcal{D}|} \sum_y \sum_{\boldsymbol{x} \in \mathcal{D}^y} D(\boldsymbol{p}_{\boldsymbol{x}} \| \boldsymbol{s}^y)}{\frac{1}{|\mathcal{D}|^2} \sum_{\boldsymbol{z} \in \mathcal{D}} \sum_{\boldsymbol{x} \in \mathcal{D}} \mathbb{1}_{\{c_{\boldsymbol{x}} \neq c_{\boldsymbol{z}}\}} D(\boldsymbol{s}^{c_{\boldsymbol{x}}} \| \boldsymbol{p}_{\boldsymbol{z}})} \equiv \tag{6}$$

$$\min_{\boldsymbol{q}^v, v \in [C]} \min_{\boldsymbol{\theta}} \left[ \frac{1}{|\mathcal{D}|} \sum_y \sum_{\boldsymbol{x} \in \mathcal{D}^y} D(\boldsymbol{p}_{\boldsymbol{x}} \| \boldsymbol{q}^y) \right] / \left[ \frac{1}{|\mathcal{D}|^2} \sum_{\boldsymbol{z} \in \mathcal{D}} \sum_{\boldsymbol{x} \in \mathcal{D}} \mathbb{1}_{\{c_{\boldsymbol{x}} \neq c_{\boldsymbol{z}}\}} \left[ H(\boldsymbol{p}_{\boldsymbol{x}}, \boldsymbol{p}_{\boldsymbol{z}}) - H(\boldsymbol{p}_{\boldsymbol{x}}, \boldsymbol{q}^{c_{\boldsymbol{z}}}) \right] \right],$$

by reformulating the single minimization problem as a double minimization problem, Equation (6) suggests an alternating algorithm, in which we use gradient descent to minimize the objective function with respect to the model's parameters $\boldsymbol{\theta}$ and centroids $\boldsymbol{q}^v, v \in [C]$ iteratively. We present PyTorch-style (Paszke et al., 2019) pseudocode for NCMI implementation in Appendix B. After training, we evaluate the NCMI-trained model on test samples by assigning each output distribution to the nearest class centroid in KL divergence. Specifically, for a model output distribution $\boldsymbol{p}$, we compute $D(\boldsymbol{p} \| \boldsymbol{q}^v)$ for each class centroid $\boldsymbol{q}^v, v \in [C]$ and predict the class with the smallest value.

## 6    EXPERIMENTS

To illustrate the effectiveness of NCMI and compare it with state-of-the-art CE-alternatives we conducted series of experiments on two widely used natural image datasets, namely CIFAR-100 (Krizhevsky et al., 2009) and ImageNet (Deng et al., 2009). The ablation studies and efficiency results are deferred to the Appendix due to space limitations.

Table 1: Top-1 validation accuracy on CIFAR-100 for models trained with NCMI and baseline methods, averaged over three random seeds. The best and second-best results are shown in **bold** and underlined, respectively.

Table 2: Top-1 and Top-5 validation accuracy on ImageNet for models trained with NCMI and baseline methods. The best and second-best results are shown in **bold** and underlined, respectively.

| CIFAR-100 | | | | |
|---|---|---|---|---|
| Model | ResNet-18 | ResNet-34 | ResNet-50 | ResNet-101 |
| CE | 75.44 | 76.42 | 76.96 | 77.39 |
| LS | 75.92 | 76.77 | 77.06 | 77.37 |
| AntiClass | 76.28 | 76.31 | 76.30 | 76.69 |
| Squentropy | 75.71 | 76.62 | 77.15 | 77.74 |
| SquareLoss | 75.10 | 76.62 | 77.15 | 77.74 |
| PolyLoss | 75.59 | 76.87 | 76.46 | 77.77 |
| SupCon | 73.00 | 74.53 | 74.88 | 75.77 |
| SupCon (large BS) | - | - | 77.04 | - |
| Focal Loss | 76.34 | 76.62 | 77.32 | 77.76 |
| NCMI (ours) | **76.45** | **76.97** | **77.50** | **77.81** |

| ImageNet | | | | |
|---|---|---|---|---|
| Method | ResNet-50 | | ResNet-101 | |
| | Top-1 | Top-5 | Top-1 | Top-5 |
| CE | 76.24 | 92.42 | 78.42 | 95.35 |
| LS | 78.37 | 94.83 | 79.10 | 96.46 |
| Focal Loss | 78.11 | 94.64 | 79.75 | 94.66 |
| SupCon | 63.78 | 86.60 | 67.43 | 90.24 |
| SupCon (large BS) | 78.70 | 94.30 | 79.33 | 94.52 |
| NCMI (ours) | **79.01** | **95.34** | **79.97** | **96.64** |

## 6.1 EXPERIMENTS ON CIFAR-100

The CIFAR-100 dataset contains 50K training and 10K test color images of resolution $32 \times 32$, which are labeled for 100 classes.

We conducted experiments on ResNets (He et al., 2016) of varying sizes. We selected ResNet-18, ResNet-34, ResNet-50 and ResNet-101 for evaluation, and we compare NCMI against to 8 benchmark methods namely, CE (Cox, 1958), LS (Szegedy et al., 2016), AntiClass (Katsikas et al., 2025), Squentropy (Hui et al., 2023), SquareLoss (Hui & Belkin, 2021), PolyLoss (Leng et al., 2022), SupCon (Khosla et al., 2020) and Focal Loss (Lin et al., 2017).

For all surrogate losses, we use an SGD optimizer with a momentum of 0.9, a learning rate of 0.1, and a weight decay of 0.0005, along with a batch size of 64. We train the model for 240 epochs, and at epochs 60, 120, and 160, we reduce the current learning rate by a factor of 10. Since SupCon typically requires large batch sizes to perform well, we report both the results from the original paper, denoted as *SupCon (large BS)*, and our reproduced results under the same setting.

The results are reported in Table 1. As seen, the models trained by NCMI outperform those trained by the baseline methods. Importantly, the improvement is consistent across various model sizes.

## 6.2 EXPERIMENTS ON IMAGENET

ImageNet (Deng et al., 2009) is a large-scale image recognition dataset that contains around 1.2M training samples and 50K validation images. We conducted experiments on two models from the ResNet family, namely ResNet-50 and ResNet-101, and evaluated NCMI's performance against CE, LS, Focal Loss, and SupCon. Similar to the CIFAR-100 setting, we use an SGD optimizer with momentum of 0.9, learning rate of 0.5, weight decay of 5e-5 and batch size of 1024, we train the models for 1000 epochs, with cosine annealing learning rate decay. For all the methods, we train the model using the image resolution of $224 \times 224$, while at evaluation, we apply a resolution of $280 \times 280$. For SupCon, we report the results under the same setting as all other methods as *SupCon* and those presented in their paper as *SupCon (large BS)*.

The results are summarized in Table 2. NCMI achieves the highest Top-1 and Top-5 accuracies; notably, under the same training setting, ResNet-50 trained on ImageNet outperforms all baselines by a large margin.

## 7 CONCLUSION

In this paper, we present a new surrogate loss for DNN-based classifiers, called normalized conditional mutual information (NCMI). We further propose a novel alternating learning algorithm to minimize the NCMI loss to train a DNN-based classifier. Extensive experimental results on CIFAR-100 and ImageNet datasets consistently show that DNN-based classifiers trained with NCMI outperform those trained using other CE-based or heuristic loss functions.

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

## A    PROOF OF THEOREM 1

We present the proof of Theorem 1 in this section.

$$
\frac{\frac{1}{|\mathcal{D}|} \sum_y \sum_{\boldsymbol{x} \in \mathcal{D}^y} D(\boldsymbol{p}_{\boldsymbol{x}} \| \boldsymbol{s}^y)}{\frac{1}{|\mathcal{D}|^2} \sum_{\boldsymbol{z} \in \mathcal{D}} \sum_{\boldsymbol{x} \in \mathcal{D}} \mathbb{1}_{\{c_{\boldsymbol{x}} \neq c_{\boldsymbol{z}}\}} D(\boldsymbol{s}^{c_{\boldsymbol{x}}} \| \boldsymbol{p}_{\boldsymbol{z}})}
$$

$$
= \frac{\frac{1}{|\mathcal{D}|} \sum_y \sum_{\boldsymbol{x} \in \mathcal{D}^y} \left[ \sum_{i=1}^k \boldsymbol{p}_{\boldsymbol{x}}[i] \ln \frac{\boldsymbol{p}_{\boldsymbol{x}}[i]}{\boldsymbol{s}^y[i]} \right]}{\frac{1}{|\mathcal{D}|^2} \sum_{\boldsymbol{z} \in \mathcal{D}} \sum_{\boldsymbol{x} \in \mathcal{D}} \mathbb{1}_{\{c_{\boldsymbol{x}} \neq c_{\boldsymbol{z}}\}} \left[ \sum_{i=1}^k \boldsymbol{s}^{c_{\boldsymbol{x}}}[i] \ln \frac{\boldsymbol{s}^{c_{\boldsymbol{x}}}[i]}{\boldsymbol{p}_{\boldsymbol{z}}[i]} \right]} \tag{7}
$$

$$
= \min_{\boldsymbol{q}^v, v \in [C]} \frac{\frac{1}{|\mathcal{D}|} \sum_y \sum_{\boldsymbol{x} \in \mathcal{D}^y} \left[ \sum_{i=1}^k \boldsymbol{p}_{\boldsymbol{x}}[i] \ln \frac{\boldsymbol{p}_{\boldsymbol{x}}[i]}{\boldsymbol{q}^y[i]} \right]}{\frac{1}{|\mathcal{D}|^2} \sum_{\boldsymbol{z} \in \mathcal{D}} \sum_{\boldsymbol{x} \in \mathcal{D}} \mathbb{1}_{\{c_{\boldsymbol{x}} \neq c_{\boldsymbol{z}}\}} \left[ \sum_{i=1}^k \boldsymbol{s}^{c_{\boldsymbol{x}}}[i] \ln \frac{\boldsymbol{q}^{c_{\boldsymbol{x}}}[i]}{\boldsymbol{p}_{\boldsymbol{z}}[i]} \right]} \tag{8}
$$

$$
= \min_{\boldsymbol{q}^v, v \in [C]} \frac{\frac{1}{|\mathcal{D}|} \sum_y \sum_{\boldsymbol{x} \in \mathcal{D}^y} D(\boldsymbol{p}_{\boldsymbol{x}} \| \boldsymbol{q}^y)}{\frac{1}{|\mathcal{D}|^2} \sum_{\boldsymbol{z} \in \mathcal{D}} \sum_{\boldsymbol{x} \in \mathcal{D}} \mathbb{1}_{\{c_{\boldsymbol{x}} \neq c_{\boldsymbol{z}}\}} \left[ H(\boldsymbol{p}_{\boldsymbol{x}}, \boldsymbol{p}_{\boldsymbol{z}}) - H(\boldsymbol{p}_{\boldsymbol{x}}, \boldsymbol{q}^{c_{\boldsymbol{z}}}) \right]} \tag{9}
$$

## B    PYTORCH-STYLE PSEUDOCODE FOR NCMI IMPLEMENTATION

We provide PyTorch-style pseudocode for the proposed NCMI training algorithm to complement the formulation in Equation (6). The listing in Algorithm 1 specifies the alternating update performed per mini-batch.

## C    TRAINING COST & TRAINING STABILITY

We quantify the efficiency on ImageNet in Table 3, reporting per-epoch wall-clock time and peak GPU memory usage. For fairness, all experiments were conducted on a server with two *AMD EPYC 7763 CPUs* and eight *NVIDIA A5000 GPUs*, using the same optimizer and data pre-processing, with batch size of 1024. As seen, compared with SupCon, NCMI takes $59.83\%$ of the graphics memory, on par with the CE and Focal Loss, while largely outperforming all the baselines in terms of classification accuracy.

NCMI also converges reliably with small batches. Figure 3 shows how batch size affects the validation accuracy of CE, SupCon, and NCMI. We train ResNet-50 on CIFAR-100 with batch sizes $\{16, 32, 64, 128, 256, 1024\}$. Across all batch sizes, NCMI exhibits robust convergence and consistently surpasses CE. In contrast, we observe that SupCon relies heavily on large batches; reducing the batch size results in a significant decline in accuracy.

**Algorithm 1** PyTorch-style pseudocode of the proposed alternating algorithm for solving the optimization problem in Equation (6).

```
                                    # model fθ; centroid ξ; momentum rate m; temperature τ;
        centroid and model optimizer optimizerξ, optimizerθ.
 1: for x, y in loader do
 2:     z ← fθ(x) − c                                              # z.shape: [B, D]
 3:     z′ ← L2NORMALIZE(z)/τ                                      # ℓ2 norm / temperature
 4:     c ← m ∗ c + (1 − m) ∗ z.mean(dim=0).detach()
                                                                   # c.shape: [1, D]
 5:     p, q ← σ^NSF(z′), σ^NSF(ξ)
                                               # p.shape: [B, D]; q.shape: [C, D]
 6:     Calculate CMI according to Equation (2)
                                                                   # CMI.shape: [B, 1]
 7:     Calculate Γ according to Equation (3)
                                                                   # Γ.shape: [B, C]
 8:     optimizerξ.zero_grad(), optimizerθ.zero_grad()
 9:     loss ← (CMI/Γ).mean()
10:     loss.backward()
11:     optimizerξ.step(), optimizerθ.step()
12: end for
```

## D  ABLATION STUDY

To understand the design choice, in this section, we evaluate the effects of the NSF and feature centering on the CIFAR-100 dataset by enabling and disabling them in all possible combinations, as shown in Table 4. Removing either component degrades performance, with the NSF having the larger impact. Replacing the NSF with a softmax head causes the model to fail to converge to a non-trivial solution.

Figure 4 further studies the effect of centering and NSF. Panel (a) shows the evolution of CMI, Γ, NCMI, and accuracy during training. Panel (b) visualizes feature clusters at epochs $\{60, 120, 200\}$ under each setting; the black crosses indicate constant-valued vectors that correspond, after $\sigma$, to the uniform distribution. Enabling centering pulls class clusters toward these reference points, thereby mitigating drift toward biased outputs. Panel (c) plots the trajectories of feature centers (and EMA-updated centers, when applicable). With centering, the centers remain stably concentrated around the constant-valued directions; without centering, they drift and collapse.

Softmax (SM) tends to produce degenerate manifolds in the t-SNE space—indicating that a few logits dominate the feature—whereas NSF suppresses overlarge entries and yields better-balanced probabilities, stabilizing optimization. Together, NSF and centering prevent single-mode collapse and avoid output distributions dominated by a few entries, leading to a more stable and reliable training process.

Table 3: Per-epoch wall-clock time and peak graph memory for ResNet-50 and ResNet-101 on ImageNet.

| | ImageNet | | | |
|---|---|---|---|---|
| | ResNet-50 | | ResNet-101 | |
| | Time ↓ | Memory ↓ | Time ↓ | Memory ↓ |
| CE | 6 mins 39 s | 102.29 Gb | 10 mins 33 s | 142.67 Gb |
| Focal Loss | 6 mins 42 s | 104.63 Gb | 10 mins 35 s | 143.42 Gb |
| SupCon | 9 mins 52 s | 180.32 Gb | 16 mins 03 s | 241.17 Gb |
| NCMI (ours) | 6 mins 44 s | 107.89 Gb | 10 mins 49 s | 148.55 Gb |

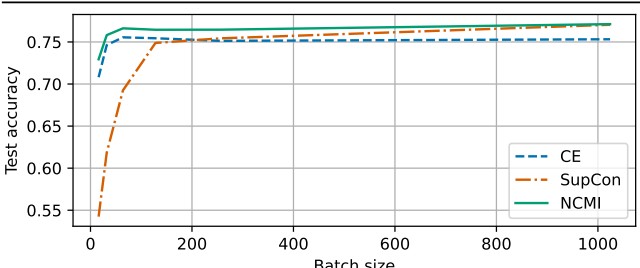

Figure 3: ResNet-50 test accuracy on CIFAR-100 as a function of batch size. We evaluate batch sizes $\{16, 32, 64, 128, 256, 1024\}$; NCMI consistently outperforms CE and SupCon across all settings.

Table 4: Component-wise ablation of NCMI on CIFAR-100. We evaluate the contribution of the NSF and the centering operation.

| | CIFAR-100 | | | |
|---|---|---|---|---|
| NSF | ✓ | ✓ | ✗ | ✗ |
| Centering | ✓ | ✗ | ✓ | ✗ |
| Accuracy | 76.45 | 74.9 | 1.72 | 5.64 |

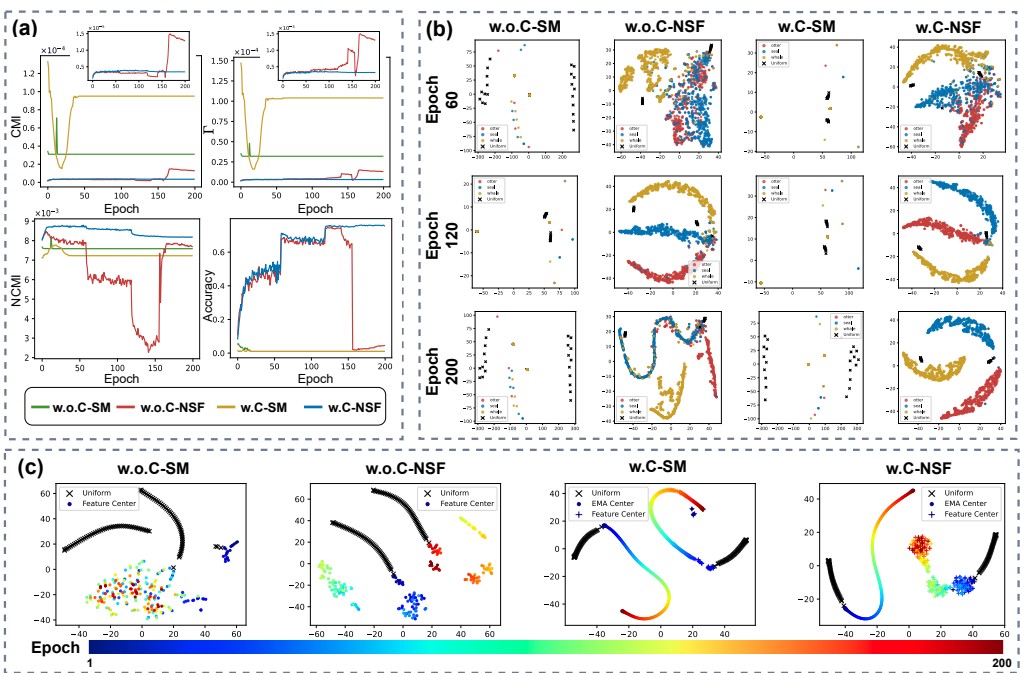

Figure 4: Ablation of feature centering and the normalized sigmoid (NSF). We ablate each component by enabling or disabling it: **w.o.C/w.C** denote without/with centering, and **SM/NSF** denote applying softmax/normalized sigmoid function. (a) Training curves of CMI, $\Gamma$, NCMI, and top-1 accuracy for ResNet-18 on CIFAR-100. (b) t-SNE of features from three randomly selected classes at epochs 60, 120, and 200; black crosses mark constant-valued vectors (all entries equal), which map via $\sigma$ to the uniform distribution. (c) t-SNE trajectories of feature centers and their EMA updates across training under all settings.

# E SCIENCE OF DL IMPROVEMENT CHALLENGE SUBMISSION

## E.1 WHAT MODEL ARE YOU TARGETING?

*Provide a summary of the problem the deep net model is designed to solve. Good summaries should outline the state of the literature, provide an overview that domain experts would consider reasonable, and cite relevant sources.*

We target standard supervised multi-class classification, where a DNN-classifier maps an input $x$ to a probability vector $p$.

The dominant training objective remains cross-entropy (CE), which optimizes agreement with pre-defined fix target distribution. Further more, empirical studies report that DNNs with compact feature clusters usually outperform those with sparse clusters (Oyallon, 2017; Papyan, 2020; Papyan et al., 2020). These insights have been analyzed under the Gaussian-mixture assumption on the feature distribution (Zarka et al., 2021). Building on this line of work, subsequent works augment CE with regularizers, Hui et al. (2023) add an $\ell_2$ penalty to the non-ground-truth entries of the predicted probability distribution, and OPL (Ranasinghe et al., 2021) explicitly clusters same-class features while enforcing orthogonality between different classes in the penultimate layer.

Another line of work improves classification accuracy by modifying CE. Focal Loss (Lin et al., 2017) down-weights well-classified examples via a power transformation so training emphasizes hard instances. PolyLoss (Leng et al., 2022) reframes standard classification losses as polynomial expansions. Hui & Belkin (2021) proposed SquareLoss , and empirically found that squared loss performs on par with or even outperforms CE on modern DNNs. Supervised contrastive learning (SupCon) (Khosla et al., 2020) pulls together same-class embeddings and pushes apart different-class embeddings, followed by a linear classifier trained on the frozen features. Further, to mitigate overconfident predictions, label smoothing (LS) (Szegedy et al., 2016) softens the one-hot targets, which can inadvertently produce compact class clusters (Müller et al., 2019). AntiClass (Katsikas et al., 2025) replaces the one-hot target with a one-cold target to mitigate neural collapse.

In contrast, this submission proposes an information-theoretic surrogate loss dubbed normalized conditional mutual information (NCMI) as a drop-in alternative to CE for training DNN classifiers. The central premise is to train classifiers by directly shaping the geometry of predicted probability vectors on the simplex: predictions should be (i) concentrated within each class and (ii) well separated across classes. The paper quantifies intra-class concentration via a conditional mutual information term $I(X; \mathcal{P}|Y)$ and quantifies inter-class separation via an aggregate divergence-to-other-class-prototypes term $\Gamma$, then defines NCMI as the ratio $I(X; \mathcal{P}|Y)/\Gamma$. Training minimizes NCMI using an alternating optimization scheme over network parameters and class "centroid" distributions, and evaluation can be performed by a nearest-centroid KL decision rule.

## E.2 HOW DO YOUR RESULTS CONTRIBUTE—OR COULD POTENTIALLY CONTRIBUTE—TO UNDERSTANDING THESE MODELS?

*What aspects of the models become better understood thanks to your work?*

Our contribution is twofold. First, we introduce an alternative training paradigm for DNN classifiers that directly shapes the geometry of the predicted probability simplex by jointly promoting intra-class concentration (predictions for the same class become consistent) and inter-class separation (class-conditional predictions become well-separated). Second, we propose a information theoretic non-likelihood surrogate loss for classifier training, enabling learning without relying on the standard cross-entropy formulation.

## E.3 HOW DO YOU EXPECT YOUR SUBMISSION TO INFLUENCE FUTURE WORK?

*Propose ways in which your insights, findings, or methodologies could shape subsequent research directions, model design choices, or scientific applications.*

The two information quantities—CMI and $\Gamma$—have clear physical meaning in information theory and information geometry, as though, NCMI may improve model interpretability by decomposing behavior into within-class concentration and between-class separation effects. Furthermore, NCMI

outperforms CE-based loss on two natural image dataset, future work can be applied it to other domains. Last but not least, developing NCMI variants to further improve the performance is a natural direction for future work.

