# OpenReview forum: "Normalized Conditional Mutual Information Surrogate Loss for Deep Learning Classifiers"
_ICLR.cc/2026/Workshop/Sci4DL — Sci4DL 2026_

### Official Review · Reviewer_GAFr · 2026-02-24

**Fit:** 3
**Significance:** 2
**Confidence:** 2

**Summary:**

This paper proposes a novel information-theoretic surrogate loss function, Normalized Conditional Mutual Information (NCMI), as an alternative to the standard Cross-Entropy (CE) loss for training deep neural classifiers. Instead of forcing model outputs to match static one-hot labels, NCMI explicitly shapes the geometry of the predicted probability simplex. It minimizes conditional mutual information (CMI) to enforce intra-class concentration while maximizing a divergence metric (Γ) to ensure inter-class separation. To overcome the computational bottleneck of global dependencies in NCMI, the authors elegantly introduce dummy centroid distributions, reformulating the objective into a GPU-friendly alternating optimization problem. Coupled with a Normalized Sigmoid Function (NSF) and feature centering to stabilize training, NCMI demonstrates highly competitive empirical results on CIFAR-100 and ImageNet, outperforming CE and contrastive baselines (like SupCon) while maintaining computational efficiency and robustness to small batch sizes.

**Strengths:**

- **Significant Paradigm Shift**: The paper offers a refreshing departure from the traditional maximum-likelihood/cross-entropy paradigm. By shifting the focus to dynamic geometric optimization on the probability simplex (intra-class compactness and inter-class separation), the work aligns perfectly with the SciForDL community's goal of understanding and improving the fundamental science of deep learning.

- **Elegant Algorithmic Translation**: The theoretical formulation is beautifully translated into practical engineering. By introducing dummy distributions to decouple the global dependencies in NCMI, the authors transform an otherwise intractable formulation into a highly efficient, GPU-friendly algorithm.

**Suggestions:**

While the theoretical premise is beautiful and the empirical results are strong, the paper’s experimental design currently conflates the underlying loss objective with heavy algorithmic stabilizers. To meet the high scientific rigor expected at SciForDL, the authors must address the following critical vulnerabilities:

**1. Disentangling the NCMI Objective from the "Algorithmic Stabilization Suite" (Critical Confounding Variables)**

The paper does not merely change the loss function; it introduces an entire optimization recipe: NCMI + NSF Activation + Feature Centering + Alternating Optimization. As shown in Table 4, removing Feature Centering causes the accuracy to plummet catastrophically (to 1.72%), proving that NCMI alone is highly ill-conditioned and suffers from trivial mode collapse without heavy regularization.

**Action Required**: The current experimental setup compares the fully-equipped "NCMI suite" against a vanilla "CE + Softmax + SGD" baseline. This is an unfair comparison. To scientifically prove that the performance gain stems from the information-theoretic NCMI objective itself, **the authors MUST provide a strong baseline where CE is trained using the same architectural stabilizers (i.e., CE + NSF + Feature Centering)**. If this fortified CE performs comparably to NCMI, it implies that the engineering tricks (smoother probability distributions and centered features) are the true drivers of success, rather than the novel loss function.

**2. Clarifying the Logical Leap: Correlation vs. Causation (Regarding Figure 2)**

Figure 2 beautifully illustrates a strong negative correlation between NCMI and accuracy across pretrained ResNet models. However, the text uses this to motivate the objective: "models with lower NCMI achieve higher accuracy. Motivated by this observation... we train the model to minimize its NCMI."

**Suggestion**: The figure itself is flawless, but the rhetorical deduction commits a correlation/causation fallacy. A low NCMI is a symptom (or a natural byproduct) of a well-separated, highly accurate model trained on CE; forcing NCMI down does not automatically guarantee learning meaningful features. The authors should carefully rephrase Section 4 to acknowledge this. NCMI should be positioned as a geometric regularization strategy inspired by this correlation, rather than a direct causal lever.

**3. Redefining the Approach as "Metric Learning on the Simplex"**

A potential point of confusion for readers is how NCMI guarantees convergence to "correct" labels if it only optimizes mutual information (which merely requires consistency, not correctness). Unlike CE, which anchors outputs to static one-hot vertices, NCMI learns dynamic centroids.

**Suggestion**: The authors should explicitly highlight early in the methodology that NCMI essentially acts as Contrastive/Metric Learning on the Probability Simplex. Explicitly stating that the target centroids are dynamically defined by the clusters themselves—meaning samples cannot concentrate on a "wrong" point because the learned centroid becomes the class identity—will resolve conceptual confusion regarding the nearest-centroid KL-divergence inference phase.

---

### Official Review · Reviewer_gWvX · 2026-02-25

**Fit:** 3
**Significance:** 2
**Confidence:** 2

**Summary:**

This paper proposes a new surrogate loss for deep neural network classification, termed Normalized Conditional Mutual Information (NCMI). Instead of optimizing cross-entropy, the method minimizes a ratio between (i) intra-class dispersion, measured via conditional mutual information, and (ii) inter-class separation, measured via a KL-based divergence term. The authors show empirically that NCMI is strongly inversely correlated with classification accuracy, and introduce an alternating optimization scheme to make the objective tractable. Experiments on CIFAR-100 and ImageNet demonstrate consistent improvements over cross-entropy and several alternative losses, with comparable computational cost. The method can be interpreted as explicitly enforcing cluster compactness and separation in the space of predicted probability distributions.

**Strengths:**

Clear conceptual motivation: The paper reframes classification through a geometric/information-theoretic lens, directly optimizing for intra-class concentration and inter-class separation rather than likelihood.

Simple and general idea: NCMI is presented as a drop-in replacement for cross-entropy and does not depend on architecture-specific modifications.

Strong empirical results: Consistent improvements across architectures (ResNets) and datasets (CIFAR-100, ImageNet), including non-trivial gains on ImageNet.

Robustness to batch size: Unlike contrastive methods, the approach appears stable even with small batch sizes.

Connection to known phenomena: The method aligns well with insights from neural collapse and representation clustering, giving it additional conceptual grounding.

**Suggestions:**

Limited theoretical justification: The key claim that accuracy is inversely proportional to NCMI is empirical; there is no formal guarantee or deeper analysis explaining why minimizing this ratio should yield optimal classifiers.

Optimization complexity: The need for alternating optimization over model parameters and class centroids introduces additional machinery compared to standard cross-entropy, which may complicate practical adoption.

Evaluation mismatch with standard practice: The use of centroid-based KL classification instead of argmax raises questions about how directly comparable the results are to standard pipelines.

Limited scope of experiments: The method is only evaluated on image classification. It is unclear whether the approach generalizes to other domains (e.g., NLP, long-tail distributions, noisy labels).

Ablation depth: While some ablations are provided (NSF, centering), more insight into the role of the normalization and the ratio structure (e.g., numerator vs denominator contributions) would strengthen the paper.

---

### Official Review · Reviewer_2u1h · 2026-02-26

**Fit:** 2
**Significance:** 2
**Confidence:** 2

**Summary:**

In this work, the authors propose a novel information-theoretic classification loss function intended as an alternative to the standard cross-entropy loss.
The core innovation lies in shifting the optimization paradigm from individual maximum likelihood estimation to the collective aggregation and segregation of features.
The paper empirically validates this approach, demonstrating that the proposed loss achieves superior classification performance compared to cross-entropy baselines across standard image datasets when evaluated with widely adopted network architectures.

**Strengths:**

This work demonstrates the potential to improve the quality of features extracted from training data when predictive distributions are used for treatment, rather than individual predictions.

**Suggestions:**

Despite the empirical superiority of the proposed method over standard cross-entropy, its theoretical limitations and principled drawbacks remain underexplored.
The authors should clarify the inherent drawbacks of their loss function and detail how the method attempts to mitigate them.
Furthermore, it would be highly beneficial to evaluate generalization quality using metrics beyond standard test accuracy.
The authors are strongly encouraged to provide a more in-depth discussion regarding the trade-offs between the individual (maximum likelihood) and distributional (collective aggregation) treatments of the predictive response.
Expanding on this analysis would elevate the paper from a predominantly empirical contribution to one that advances our fundamental understanding of these two distinct learning paradigms.

---

### Meta-Review · Area_Chair_8P8B · 2026-02-28

**Recommendation:** Accept

**Metareview:**

All reviewers state the paper is making an interesting contribution by proposing a more geometric loss for classification. It demonstrates empirical advantage over cross entropy loss. The main drawback is the lack of theoretical justification for the empirical results.

The paper is a good fit for the workshop.

---

### Decision · Program_Chairs · 2026-03-02

Accept